# Anti-Inflammatory Effects of Synthetic Peptides Based on Glucocorticoid-Induced Leucine Zipper (GILZ) Protein for the Treatment of Inflammatory Bowel Diseases (IBDs)

**DOI:** 10.3390/cells12182294

**Published:** 2023-09-16

**Authors:** Musetta Paglialunga, Sara Flamini, Raffaele Contini, Marta Febo, Erika Ricci, Simona Ronchetti, Oxana Bereshchenko, Graziella Migliorati, Carlo Riccardi, Stefano Bruscoli

**Affiliations:** 1Department of Medicine and Surgery, Section of Pharmacology, University of Perugia, 06132 Perugia, Italy; musetta.paglialunga@gmail.com (M.P.); sarafla@hotmail.it (S.F.); raffaele.contini@studenti.unipg.it (R.C.); marta.febo@studenti.unipg.it (M.F.); erika.ricci.1985@gmail.com (E.R.); simona.ronchetti@unipg.it (S.R.); graziella.migliorati@unipg.it (G.M.); riccardi@unipg.it (C.R.); 2Department of Philosophy, Social Sciences and Education, University of Perugia, 06123 Perugia, Italy; oxana.bereshchenko@unipg.it

**Keywords:** glucocorticoids, GILZ, inflammation, NF-κB, IBDs

## Abstract

Glucocorticoids (GCs) are commonly used to treat autoimmune and inflammatory diseases, but their clinical effects and long-term use can lead to serious side effects. New drugs that can replace GCs are needed. Glucocorticoid-induced leucine zipper (GILZ) is induced by GCs and mediates many of their anti-inflammatory effects, such as inhibiting the pro-inflammatory molecule NF-κB. The GILZ C-terminal domain (PER region) is responsible for GILZ/p65NF-κB interaction and consequent inhibition of its transcriptional activity. A set of five short peptides spanning different parts of the PER region of GILZ protein was designed, and their anti-inflammatory activity was tested, both in vitro and in vivo. We tested the biological activity of GILZ peptides in human lymphocytic and monocytic cell lines to evaluate their inhibitory effect on the NF-κB-dependent expression of pro-inflammatory cytokines. Among the tested peptides, the peptide named PEP-1 demonstrated the highest efficacy in inhibiting cell activation in vitro. Subsequently, PEP-1 was further evaluated in two in vivo experimental colitis models (chemically induced by DNBS administration and spontaneous colitis induced in IL-10 knock-out (KO) mice (to assess its effectiveness in counteracting inflammation. Results show that PEP-1 reduced disease severity in both colitis models associated with reduced NF-κB pro-inflammatory activity in colon lamina propria lymphocytes. This study explored GILZ-based ‘small peptides’ potential efficacy in decreasing lymphocyte activation and inflammation associated with experimental inflammatory bowel diseases (IBDs). Small peptides have several advantages over the entire protein, including higher selectivity, better stability, and bioavailability profile, and are easy to synthesize and cost-effective. Thus, identifying active GILZ peptides could represent a new class of drugs for treating IBD patients.

## 1. Introduction

Current anti-inflammatory pharmacological therapies include small molecules and biotechnological drugs; however, inflammatory diseases, including inflammatory bowel diseases (IBDs), remain incurable. Thus, novel therapies for such recurrent and life-threatening conditions are urgently needed. 

Glucocorticoids (GCs) are endogenous hormones produced and released by the adrenal cortex. Endogenous and synthetic GCs have numerous physiological and pharmacological effects, influencing the function of most cells in the body.

At pharmacological doses, GCs have anti-inflammatory and immunosuppressive activities involving nearly all players of the inflammatory response, including T lymphocytes and macrophages [1,2,3]. Exploiting the GCs’ physiologic mechanisms, GCs are widely used for therapy. Still today, many diseases require therapy with GCs, such as autoimmune diseases and chronic inflammatory diseases, including most allergic reactions, asthma, neurological and hematological disorders, and IBDs [4,5,6]. However, despite GCs’ potent anti-inflammatory activities, their clinical effects are transitory; the disease recurs on tapering the drug, while chronic use of GCs is accompanied by serious side effects that cause therapy discontinuation [2,7,8]. Therefore, new drugs substituting GCs may provide critical aid in the therapy of inflammatory diseases. 

NF-κB proteins are a family of ubiquitously expressed transcription factors that play an essential role in most immune and inflammatory responses and are involved in the control of many cellular processes, such as activation, cell growth, and apoptosis. Although NF-κB is known to have beneficial functions in maintaining normal physiological processes, a deregulated activation of NF-κB has been linked to the development and progression of various disorders, including inflammatory diseases such as IBDs [9,10,11]. Due to NF-κB implication in inflammatory diseases [12] and the fact that mouse models of IBDs and other inflammatory diseases respond positively to NF-κB inhibitors [13], significant attention has been focused on the development of anti-inflammatory drugs targeting NF-κB [10,14]. Importantly, GCs are potent inhibitors of NF-κB transcriptional activity and NF-κB-mediated inflammatory responses, and interference with NF-κB signaling pathways is one of the most important mechanisms underlying the anti-inflammatory effects of GCs. The inhibition of NF-κB is one of the mechanisms through which GCs induce apoptosis in cells of the hematopoietic system, such as monocytes, macrophages, and T lymphocytes, all involved in the inflammatory response [15,16]. 

Since GC-induced effects are mostly due to the modulation of target gene expression, we initiated studies aimed at identifying proteins induced by GC treatment that mediates the anti-inflammatory but not the side effects of GCs. In this context, we identified the glucocorticoid-induced leucine zipper (GILZ) protein as a mediator of the anti-inflammatory effects of GCs. GILZ (or tsc22d3), both murine and human, is a gene rapidly induced by GCs [17]. GILZ can modulate proliferation/activation, survival (mainly apoptosis), and differentiation of different cells, including lymphocytes and macrophages. Murine GILZ encodes a 137-aa leucine zipper (LZ) protein (7), and human GILZ, encoding for a 135-aa protein, is almost identical to its murine homolog [18]. Structurally, GILZ has a complex composition that consists of four distinctive domains: an N-terminal domain, which spans residues 1–60 amino acids (aa); a TSC-box (61–75 aa), highly conserved in the tsc22d family proteins [19,20]; a central leucine zipper (LZ) domain, found in residues 76–97; and a C-terminal domain, featuring a proline (P)/glutamic acid (E) rich region, also known as the PER region, with residues ranging from 98 to 137 [21].

During recent years, a lot of experimental evidence supports the role of GILZ as a mediator of the anti-inflammatory activity of GCs [22,23,24,25,26]. GILZ is involved in the modulation of the same immune responses and inflammation-related signaling pathways implicated in GC-induced anti-inflammatory and immunosuppressive activities. Of note, we found that activated macrophages in the granulomas of patients with active Crohn’s disease do not express GILZ [23]. Moreover, we also found that while GILZ KO mice are more susceptible to spontaneous and induced colitis, GILZ overexpression in GILZ-transgenic (TG) mice or administration of full-length GILZ recombinant fusion protein counters IBDs [27,28]. In vivo administration of TAT–GILZ fusion protein is efficacious in alleviating intestinal inflammation in different experimental models of IBDs [27,29]. We have also shown that the TAT–GILZ fusion protein efficiently promotes the resolution of neutrophilic inflammation induced by LPS [25]. Several other studies indicate that GILZ over-expression, as well as in vivo delivery of the GILZ fusion protein, can achieve many of the anti-inflammatory effects of GCs, including inhibition of T-cell activation, proliferation and differentiation, and macrophage activation. To note, no apparent toxic effects have been ever observed in TAT–GILZ- or GILZ-peptide-treated mice [30,31,32,33,34,35].

A body of accumulating experimental evidence from our and other laboratories suggests that GILZ inhibits NF-κB transcriptional activities in different tissues and cell types, including lymphoid and myeloid cells. Initial observation showed that GILZ directly binds to and inhibits NF-κB activation, transcriptional activity, and nuclear translocation [36,37]. GILZ mimics the effect of GCs on the differentiation of T-cells since GILZ over-expression in T-cells in GILZ transgenic (TG) mice induces down-regulation of the T helper (Th1) and up-regulation of Th-2 response [28]. GILZ overexpression in T-cells is associated with inhibition of NF-κB nuclear translocation in CD4+ T lymphocytes of intestinal lamina propria (LP) and a decreased susceptibility to Th1-mediated colitis [27]. To note, GILZ over-expression prevents the NF-κB-dependent induction of pro-inflammatory target genes in T lymphocytes, including inhibition of IL-2 production [21,23].

Other experimental approaches, using several in vitro and in vivo models, further supported the link between the anti-inflammatory activity of GILZ and the inhibition of NF-κB in T lymphocytes [21,36]. Furthermore, it was demonstrated that inhibition of macrophage activation by GCs is achieved by GILZ-mediated inhibition of NF-κB [23,38,39]. GILZ is constitutively expressed in mouse and human macrophages and human monocytes and is up-regulated by GCs [18,23]. In the THP-1 human macrophage cell line, GILZ binds to p65 and inhibits the expression of co-stimulatory molecules CD80 and CD86 and the production of RANTES and MIP-1 chemokines, thus mimicking the inhibiting effect of GCs treatment on macrophages [23]. Furthermore, loss of GILZ in aging macrophages in mice increases susceptibility to tissue inflammation [40]. 

We previously identified the specific amino acidic region of GILZ important for GILZ NF-κB interaction: the C-terminal part of GILZ protein contains five PXXP motifs that play a crucial role in GILZ’s interaction with NF-κB [21]. In the present work, we propose to identify novel anti-inflammatory GILZ-based peptides, designed on the NF-κB binding region of GILZ protein, responsible for the inhibitory action of GILZ on NF-κB transcriptional activity and its inflammatory effects.

## 2. Materials and Methods

### 2.1. Cell Culture

Jurkat human T lymphocytic leukemia cell line was purchased from the American Type Culture Collection, (ATCC, Manassas, VA, USA). Jurkat cells were cultivated in suspension in Advanced RPMI 1640 (Gibco/Invitrogen, Paisley, UK) supplemented with 10% fetal bovine serum (FBS) (Gibco/Invitrogen, Paisley, UK) and 10 mM Hepes, 1 mM of sodium pyruvate, and 1% Pen/Strep 5000 U/mL (Gibco/Invitrogen, Paisley, UK).

THP-1 human monocytic leukemia cell line (ATCC, Manassas, VA, USA) was cultivated in Advanced RPMI 1640 (Gibco/Invitrogen, Paisley, UK) supplemented with 10% fetal bovine serum (Gibco/Invitrogen, Paisley, UK) and 1% Pen/Strep 5000 U/mL (Gibco/Invitrogen, Paisley, UK). Cell cultures were maintained at 37 °C with 5% CO_2_ and examined daily under a light microscope for quality control. 

### 2.2. Peptide Synthesis

GILZ lyophilized peptides were purchased from ProteoGenix (Schiltigheim, France). They were synthesized using solid-phase peptide synthesis (SPPS), attaching peptide chains to a polymeric solid support in a single reaction (one-pot synthesis). Each peptide carried out acetylation as N-terminal modification, and crude peptides were purified to >95% purity using high-performance liquid chromatography (HPLC) after being reconstituted in Acetonitrile (ACN) at ACN: H_2_O = 1:3 according to manufacturer’s instructions.

GILZ scrambled peptide sequence was obtained using an analytical tool, “PeptideNexus,” and the sequence was compared to protein query sequences and databases using the Basic Local Alignment Search Tool, BLAST (NCBI). 

Peptides shared an initial aminoacidic sequence, YGRKKRRQRRR, corresponding to the sequence derived from the trans-activator of transcription (TAT) protein of the human immunodeficiency virus, known as a cell-penetrating peptide, which was used to deliver GILZ peptides inside the cell to carry out their biological actions [41]. In Figure 1C, the TAT sequence precedes each GILZ peptide sequence.

### 2.3. In Vitro Cell Activation

Jurkat or THP-1 cells were seeded at a concentration of 1 x 10^6^/mL on 12-well 3.5 cm^2^ cell culture plates (ThermoFisher Scientific, Waltham, MA, USA) and then treated with TAT–GILZ peptides.

Jurkat cells were treated with each GILZ peptide for 1 h at a concentration of 0.1 μM, then activated with Phorbol Myristate Acetate (PMA) 50 ng/mL and Ionomycin 1.5 μM for 6 h at 37 °C with 5% CO_2_. Jurkat cells were collected and pelleted up by centrifuging at 4 °C for 5 min at 300× *g* (1500 rpm) in collection tubes and stored for further analyses: an aliquot was homogenized with 350 μL of RNA-Xpress^TM^ (HIMEDIA, Mumbai, India) reagent for RNA extraction and qPCR analysis, a second aliquot was used to study NF-κB phosphorylation by flow cytometry.

Conversely, THP-1 cells were treated with GILZ PEP-1 peptide for 1 h at a concentration of 5 μM and subsequently stimulated with LPS 2 μg/mL for 6 h at 37 °C with 5% CO_2_. After collection, THP-1 cells were pelleted up and homogenized with 350 μL of RNA-Xpress^TM^ reagent for qPCR analysis. 

### 2.4. Mice

Il10 deficient mice (IL-10KO mice), spontaneously develop chronic enterocolitis that resembles human inflammatory bowel disease (IBD), particularly Crohn’s disease [42]. IL-10 KO mice were purchased from Jackson Laboratories (Charles River, Milan, Italy). C57BL/6JOlaHsd males were purchased from Envigo (Udine, Italy).

Animals were housed in a controlled environment (12 h dark/light cycle) and provided with standard rodent chow and water ad libitum. Animal care complied with national ethical guidelines (Italian Ministry of Health; DL 26, 4 March 2014) and the guidelines from Directive 2010/63/EU of the European Parliament. 

IL-KO mice of 7 weeks of age were weighed daily during the entire period of treatment with peptides to calculate the percentage of body weight loss and the total score of disease. Score disease was evaluated as described in [32] according to the following criteria: 0 = no damage, 1 = localized hyperemia without ulcers, 2 = linear ulcers with no significant inflammation, 3 = linear ulcers with localized inflammation, and 4 = two or more significant areas of inflammation and ulceration extending more than 1 cm along the length of the colon. Following mice sacrifice, colons were collected to estimate the weight/length ratio and used for histological examination of sub-mucosal tissues. Infiltrated leukocytes in lamina propria were isolated and counted; afterward were used for flow cytometry and immunofluorescence analyses.

### 2.5. DNBS-Induced Colitis

Colitis was induced in 10 to 14-week-old C57BL/6JOlaHsd males (Envigo, Udine, Italy). They were anesthetized with sodium thiopental (30 mg/kg) and xylazine (10 mg/kg). A 2 mg dose of 2,4-Dinitrobenzene sulfonic acid (DNBS, Sigma–Aldrich, St. Louis, MO, USA) or vehicle 50% ethanol (SHAM) was injected intrarectally. To evaluate whether GILZ peptide PEP-1 administration protected against the development of colitis, DNBS-treated mice were randomized to receive no treatment (DNBS alone) or PEP-1 treatment. Daily assessments of body weight, stool consistency, and rectal bleeding were made, and the total score disease was determined as previously described. Mice colons were collected to calculate the weight/length ratio and processed to quantify infiltrating cells in lamina propria.

### 2.6. Tissue Histology

Tissues were fixed for 24 h in paraformaldehyde (PFA) solution (4% in PBS 0.1 M), dehydrated in graded ethanol, embedded in paraffin (Paraplast, Sherwood Medical, Mahwah, NJ, USA), and sectioned. Tissue sections (thickness, 7 μm) were deparaffinized with xylene, stained with hematoxylin and eosin (H&E), and were observed by light microscopy (LEICA DM 2000 combined with a LEICA ICC50 HD camera, Wetzlar, Germany).

### 2.7. Leukocyte Isolation from Colon Lamina Propria

Leukocytes present in the lamina propria (LP) of the colons of colitic mice were isolated, as described previously [32]. Briefly, colons isolated from mice were washed and incubated for 15 min at 37 °C with dissociation buffer composed of Ca/Mag-free RPMI containing 5 mM EDTA, 10 mM Hepes, and 1 mM dithiothreitol (DTT) and subsequently washed with complete RPMI containing 10 mM HEPES and 10 mM 10% fetal bovine serum (FBS). After this procedure, fragments were then incubated for 20 min at 37 °C with reheated digestion buffer (RPMI complete containing 0.5 mg/mL collagenase IV (Sigma–Aldrich, MO, USA) and 0.5 mg/mL DNaseI). After a wash, cells were exposed to density gradient centrifugation in 40/80% Percoll, and LP cells collected from the interface were washed with PBS containing 2% FBS. Isolated LP cells resuspended in PBS were used for flow cytometric and immunofluorescence analyses.

### 2.8. RNA Isolation and qPCR

RNA was isolated using an RNA-Xpress^TM^ reagent according to the manufacturer’s protocol, and the reverse transcription reaction foresaw the incubation of 1 μg of total RNA from each sample with All-in-One 5X RT Master Mix (Abcam, Cambridge, UK). qPCR was performed using the QuantStudio 1 Real-Time PCR System (Applied Biosystems, Foster City, CA,USA) and TaqMan Gene Expression Master Mix (Applied Biosystems, CA, USA). The qPCR TaqMan probes (Applied Biosystems, Foster City, CA, USA) were as follows: Hs00174114 m1_IL2; Hs00907777_m1 IL2RA; IL-1β: Hs01555410_m1; TNF: Hs00174128_m1; and IL-6: Hs00174131_m1. β-actin was used as a housekeeping gene, all experiments were carried out in triplicate, and the ΔΔCt method was used to quantify the relative expression of all the cytokines tested. 

### 2.9. Flow Cytometry

Jurkat cells and isolated leukocytes from colon LP were collected for flow cytometric analysis. Cells were fixated for 12 min in the dark with a solution composed of 100 μL of PBS 2% FBS and 100 μL PFA 4%. Afterward, cells were pelleted up (5 min at 500 rcf) and resuspended in a blocking/permeabilization buffer for 30 min composed of PBS 2% FBS and saponin 0.2% to allow perforation of membranes. After centrifugation, cells were incubated with an antibody staining mix formed by phospho-p65/NF-kB monoclonal antibody (Cell Signaling Technology, Beverly, MA, USA) and PBS 2% FBS + saponin 0.2% in the dark for a minimum of 30 min. Data were analyzed with the ATTUNE NxT three-laser standard setup (Life Technologies, Carlsbad, CA, USA) and the FlowJo software v10 (Tree Star, Woodburn, OR, USA) [43].

### 2.10. Immunofluorescence

LP cells infiltrated in the colon were incubated in the presence or absence of plated-bound anti-CD3ε (2 μg/mL) for 15 min. After treatment, cells followed the paraformaldehyde-saponin procedure for immunofluorescence staining. After vigorous washing in PBS with 1% HEPES, cells were fixed in 4% formaldehyde on ice for 20 min, washed once more, and then incubated in a blocking buffer at 4 °C for 1 h (PBS with 3% bovine serum albumin and 1% glycine).

Isolated cells were cytospun onto glass slides for 4′ at 400 rpm. Spotted cells were washed in phosphate-buffered saline (PBS) and then were permeabilized by incubation in paraformaldehyde 4% for 20 min at r.t. After washing three times in PBS, cells were blocked with blocking buffer containing PBS, 1% BSA, and 10% horse serum, for 1 h at room temperature. The slides were then incubated with anti-p65 NF-κB antibody (Santa Cruz, Santa Cruz, CA, USA) o.n. at 4 °C. After 3 washes with PBS/Tween 0.1%, the slides were then incubated for 1 h with a secondary anti-rabbit Alexa Fluor 555-conjugated antibody. Nuclei were stained with diamidino-2-phenylindole (DAPI). After washing, slides were mounted with cover glass and analyzed using a Zeiss Axioplan fluorescence microscope. 

The quantification of NF-κB/p65 nuclear translocation was calculated by measuring the NF-κB localization on the cytoplasmic region and normalized by counting the total number of cells stained with DAPI.

### 2.11. Statistical Analysis

Prism 7.0 (GraphPad, Boston, MA, USA) was used for all statistical analyses. The results in the figures are representative of at least 3 experiments performed on different experimental days. The two-tailed unpaired Student’s *t*-test was used for statistical comparison with the controls (* *p* < 0.05; ** *p* < 0.005; *** *p* < 0.001; **** *p* < 0.0001).

## 3. Results

### 3.1. Anti-Inflammatory Activity of GILZ Peptides in Human Cell Lines

It has been previously demonstrated that GILZ recombinant protein has anti-inflammatory properties [19,44]. It has been shown that a C-terminal domain of GILZ protein, named PER region, is necessary for GILZ–NF-κB interaction [21], as indicated in Figure 1A. We generated five peptides, named PEP-1-5: PEP-1 is the longest peptide with a stretch of 23 aa that spans the entire PER region and is responsible for the interaction with NK-κB PER (aa 115–137), while the other peptides (PEP-2, PEP-3, PEP-4 and PEP-5) are smaller (nine to ten residues in length) and drawn on different portions of the PER region, as indicated in Figure 1C.

To assess the potential efficacy of GILZ peptides as anti-inflammatory molecules, we first tested their ability to inhibit cell activation. Jurkat cells were activated or not with PMA and ionomycin (PMA + IONO), and the mRNA expression of the pro-inflammatory cytokine, interleukin-2 (IL-2), and its receptor, interleukin-2 receptor alpha (IL-2Rα), was analyzed by qPCR in cells pre-treated for one hour with PEP-1-5 or with scramble peptide as a control. As shown in Figure 2A, pre-treatment before activation of Jurkat cells with PEP-1 (column 3) and PEP-2 (column 4) caused a significantly decreased induction of IL-2 mRNA expression levels compared to the controls (vehicle or scramble-treated cells, columns 1 and 2, respectively) while PEP-5 peptide did not show any significant effect (Figure 2A, columns 7). Surprisingly, activated Jurkat cells treated with PEP-3 and PEP-4 peptides show an increase in IL-2 activation compared to controls (Figure 2A, columns 5 and 6 vs. columns 1 and 2). This pattern of IL-2 expression similarly corresponds to the IL-2Rα expression levels upon pre-treatment with various peptides, with PEP-1 administration that led to a significantly decreased mRNA expression levels of IL-2Rα compared to controls (Figure 2B, column 3 vs. columns 1 and 2), and with PEP-3 treatment that enhances expression of IL-2Rα (Figure 2B, column 5 vs. columns 1 and 2), while treatment with other peptides did not show significant variation compared to controls.

Collectively, PEP-1 appeared to be the best peptide to counteract the induction of IL-2 and IL-2R expression upon T-cell activation.

### 3.2. Evaluation of NF-κB Nuclear Translocation by Flow Cytometry in Jurkat Cells 

IL-2 is one of the most important pro-inflammatory cytokines produced by activated T-cells [45]. Its expression is regulated by the NF-κB transcription factor following its activation and nuclear translocation [46]. NF-κB nuclear translocation in stimulated Jurkat cells treated or not with PEP-1 was analyzed by flow cytometry. We evaluated the efficacy of PEP-1 peptide in inhibiting NF-κB activation by staining PMA + IONO-activated Jurkat cells with phospho-NF-κB/p65 (pNF-κB) antibody, which detects activated NF-κB forms [47].

Figure 3 shows the effect of the treatment of Jurkat cells with different concentrations of PEP-1, with the highest concentration tested (5 μM) being able to significantly decrease the frequency of pNF-κB/p65-positive cells compared to control cells treated with scramble peptide, indicating that PEP-1 peptide inhibits NF-κB phosphorylation and activation.

### 3.3. Anti-Inflammatory Activity of PEP-1 in Human Monocytes

Since PEP-1 showed significant effects in Jurkat cells, we tested its anti-inflammatory activity in a second cell line, the monocytic line THP-1. We have evaluated the effect of PEP-1 treatment on the expression levels of pro-inflammatory cytokines and NF-κB target genes IL-1β, TNF-α, and IL-6 in THP-1 cells activated with Lipopolysaccharide (LPS).

One hour pre-treatment with PEP-1 peptide did not affect the induction of the mRNA expression of IL-1β by LPS (Figure 4A) but significantly reduced the levels of the mRNA induction of TNF-α (Figure 4B) and IL-6 (Figure 4C) in activated THP-1, compared to controls (scramble-treated cells). 

Altogether, these results showed an anti-inflammatory activity of PEP-1 peptide also in the human monocytic cell line THP-1.

### 3.4. PEP-1 Peptide Administration Ameliorates Clinical Signs in Colitic IL-10KO Mice

Based on the in vitro data acquired, it was observed that the peptide PEP-1 exhibited the most significant inhibitory effect on pro-inflammatory cytokine IL-2 transcription. Consequently, further investigation was conducted by testing this peptide in a genetic model of spontaneous colitis (IL-10KO mice) to evaluate its therapeutic potential in vivo. IL-10KO mice spontaneously developed Th1-dependent chronic enterocolitis at 8–10 weeks of age [42]. To evaluate the therapeutic activity of the PEP-1 peptide, 5 μg/mouse of the peptide was administrated intra-peritoneally, three times a week, starting at the age of seven weeks, one week before the onset of the established disease [42]. 

Mice were sacrificed after 6 weeks of treatment with PEP-1 peptide or scramble peptide (control). We observed that PEP-1 treatment inhibits the severity of colitis in IL-10KO mice, as displayed by a significant recovery in body weight loss compared to the control (Figure 5A). Daily evaluation of stool consistency and rectal bleeding, represented by a score disease in Figure 5B, confirms the efficacy of PEP-1 treatment in slowing down the disease susceptibility and the progression of colitis in IL-10KO mice compared to control, treated with scramble peptide (Figure 5B).

In addition, differences in colon weight to length ratio, a parameter useful to understand the severity of tissue edema, were evident and significantly reduced in PEP-1-treated mice in comparison to scramble-treated mice (Figure 5C), further indicating that PEP-1 treatment was efficacious and ameliorated signs of colitis in IL-10KO mice.

### 3.5. PEP-1 Peptide Treatment Counteracts Leukocytes Infiltration and Prevents NF-κB Nuclear Translocation in Colon Lamina Propria (LP) of IL-10 KO mice

After mice sacrifice, morphological analysis of the colons of IL-10KO mice by H&E staining indicated that colons of PEP-1-treated mice showed reduced inflammatory lesions and less infiltration of immune cells in the sub-mucosal tissues, indicated by white arrows in Figure 6A in the scramble-treated mice. Leukocytes isolated from LP of colitic mice were counted, and a substantial decrease in the number of infiltrating cells was observed in colon LP upon PEP-1 treatment, compared to scramble-treated mice (Figure 6B), thus confirming the efficacy of PEP-1 in the reduction of the severity of colitis in IL-10KO mice.

Furthermore, to assess the levels of NF-κB nuclear translocation and activation in the leukocytes infiltrating the colon LP, we cultured leukocytes isolated from LP of the colons of IL-10KO mice, stimulated for 15 min with plated-bound anti-CD3 (2 μg/mL) to activate NF-κB nuclear translocation, and treated with PEP-1 or scramble peptide. NF-κB activation was assessed by staining with Ser 536 phospho-NF-κB antibody (pNF-κB), which revealed activated NF-κB form [47]. As reported in Figure 7, the pNF-κB positive LP cells treated with PEP-1 substantially reduced the frequency of pNF-κB positive cells compared to the scramble control. 

Moreover, to investigate the inhibition of NF-κB nuclear translocation, we assessed the NF-κB nuclear translocation in LP cells of the colon of colitic IL-10KO mice treated or not with PEP-1 or scramble peptide by immunofluorescence. Results indicate that LP cells of PEP-1-treated mice showed a significant reduction of NF-κB nuclear translocation compared to scramble-treated controls (Figure 8).

### 3.6. PEP-1 Protect against the Development of DNBS-Induced Colitis in Mice

To confirm the anti-inflammatory activity of the PEP-1 peptide in vivo, we evaluated the efficacy of PEP-1 in another model of acute colitis chemically induced by DNBS administration [48]. The severity of the DNBS-induced colitis was monitored daily and scored by a disease index score, which incorporated several parameters, including body weight loss, stool consistency, and rectal bleeding. Results of the follow-up represented in Figure 9A show a decreased score disease in PEP-1-treated mice compared to controls. After sacrifice, colons from colitic mice or from Sham controls were collected and examined for macroscopic signs of disease (shortening, bleeding, and stool consistency, Figure 9B), providing clear evidence of the differences in the inflammatory signs in the colons of PEP-1-treated compared to those of not-treated colitic mice. There was a slight decrease in the colon weight-to-length ratio in the PEP-1 treated group compared to the control group (Figure 9C). Moreover, cells were isolated from the colons of LP of diseased mice, treated or not with PEP-1, and the total number of leukocytes infiltrated in colon LP was significantly decreased in PEP-1-treated mice in contrast with the controls (Figure 9D), indicating that the inflammatory process was partially ameliorated by PEP-1 administration during DNBS-induced colitis.

## 4. Discussion

GCs are potent anti-inflammatory and immunosuppressive agents, still in use as first-line drugs to treat many immune cell-mediated diseases, including autoimmune and chronic inflammatory diseases. Systemic, or more recently also topical, administration of GCs remains the basic treatment for moderate to severe IBD, but their use in chronic conditions is limited by several important adverse drug effects [2,5,49]. 

A major part of the physiological and pharmacological effects of natural and synthetic GCs involves the activation of glucocorticoid receptor (GR), a transcription factor belonging to the nuclear receptor superfamily [15,50,51]. Upon ligand binding, GR migrates to the nucleus and binds to GC-response elements in the promoters of target genes, leading to positive or negative effects in gene expression and consequent protein synthesis. Almost all the effects of GCs, therapeutic and unwanted, are mediated by GR transcriptional activity. Long-lasting efforts to separate the beneficial from harmful gene activation by modulating the GR activity have not yielded any success, and the GCs toxicity is still a big issue in clinical practice. For these reasons, it is very important to find other ways to separate their benefits from adverse effects. An approach is to exploit the therapeutic potential of the downstream GC-induced effectors molecules, such as GILZ. Several lines of evidence demonstrate that GILZ, transcriptionally up-regulated by GCs, is a mediator of many anti-inflammatory effects of GCs [22,23,24,25,26,44]. Moreover, higher levels of GILZ expression have been linked to resistance to LPS-induced lethality in a particular inbred mouse strain SPRET/Ei [35], and transgenic mice overexpressing GILZ displayed lower levels of colonic inflammation [28]. 

Therefore, the development of strategies to achieve increased levels of GILZ protein that do not involve treatments with GCs represents a way to restrain inflammation potentially applicable to various inflammatory diseases. The delivery of GILZ protein or GILZ-derived protein sequence has been tested in experimental models using cells and animal models [25,27,29,30,31,32,33,34,35]. This approach was proven successful in inhibiting inflammation in experimental models, including ours and several other groups. At first, the delivery of bacterially expressed and purified full-length GILZ protein, fused in-frame with the short peptide corresponding to the sequence of the HIV-encoded TAT protein that ensures membrane penetration, demonstrated its anti-inflammatory efficacy in different experimental models of IBDs [27,29]. Moreover, a shorter GILZ protein sequence-derived peptide was shown to be effective in suppressing neuroinflammation in the mouse model of induced multiple sclerosis, demonstrating the feasibility of the experimental approach of using GILZ-based peptide as a pharmacologic tool in an in vivo model of multiple sclerosis [52]. More recently, another study demonstrated the therapeutic efficacy of an in vivo delivery of a GILZ-based peptide in the prevention of degenerative retina in rats [53]. 

In the present work, we tested the therapeutic efficacy of GILZ-derived peptides in alleviating the colitis symptoms and reducing the activity of NF-κB, which represents a central factor in inflammatory diseases, including IBDs [12,54]. We have previously characterized the GILZ/NF-κB interaction in T lymphocytes and macrophages [21,25,36,38]. In particular, we have shown that the first 29 amino acids of the proline-rich region (PER, amino acids 98–127), flanking the LZ domain, are important for NF-κB functional inhibition and, in particular, that the PXXP motifs (present in the C-terminal PER domain) are essential for GILZ/NF-κB binding [21]. To note, the peptide that was used for in vivo suppression of experimental autoimmune encephalomyelitis (EAE) corresponded to the C-terminal portion of the GILZ protein [52]. Since GILZ interacts with NF-κB via its C-terminal PER domain, we first refined the functional characterization of the C-terminal part of GILZ protein, with the generation of five overlapping peptides of different length, spanning the C-terminal region of GILZ (amino acids 115–137, Figure 1), potentially inhibiting NF-κB activity. 

To define the region and the peptide sequence able to achieve the suppression of NF-κB pro-inflammatory effects, both in vitro and in vivo assays were performed. We have used two different human cell lines of T-cells (Jurkat) and monocytic (THP-1) origin to assess the immunomodulatory activity of the five peptides in two different immune lineages. 

The results obtained in Jurkat T-cells, which were activated with PMA and ionomycin to induce NF-κB-dependent up-regulation of IL-2 and IL-2Rα expression, demonstrated that the five peptides did not behave similarly in terms of their ability to suppress the expression of the pro-inflammatory cytokine IL-2. Compared to others, the PEP-1 peptide showed the most pronounced ability to prevent IL-2 activation. Its anti-inflammatory activity was then tested and confirmed in the THP-1 monocytic cell line. These cells, like the body macrophages, express pro-inflammatory cytokines TNFα and IL-6, target genes of NF-κB transcriptional activity upon cell activation [55]. The treatment of these cells with PEP-1 reduced the activation of the mRNA expression levels of TNFα and IL-6 upon activation with LPS. These data demonstrated that PEP-1 peptide possesses an anti-inflammatory activity in vitro and could be adopted for in vivo testing. 

Of note, the in vitro experiments performed in Jurkat cells showed that PEP-3, PEP-4, and PEP-5 peptides were not able to suppress the IL-2 induced by PMA and ionomycin; the treatment with these peptides was associated with a slight (in case of PEP-5) or up to 2-fold (in case of PEP-3) increase of IL-2 expression compared to controls (Figure 2A). Similar data were obtained assessing IL-2Rα expression levels in activated Jurkat cells (Figure 2B). These functional differences in the selected peptide might derive from the differences in amino acid sequences of the C-terminal region of the GILZ protein (Figure 1). These molecules could have a mimetic GILZ effect, as in the case of PEP-1, which consists of the longest sequence (23 amino acids, compared to 9 or 10 amino acids of other peptides) and is more likely to mimic the effects of the whole GILZ protein, while the smaller peptides could interfere with the effects of the endogenous protein, perhaps still binding to the p65/NF-κB subunit, without having an inhibitory effect against NF-κB itself. However, these hypotheses, although interesting and potentially useful for possible applications on the enhancement of the NF-κB pathway, need further investigation and new experiments demonstrating this activity.

Based on the in vitro results, and coherently with the aim of the study on the definition of GILZ mimetic molecules with anti-inflammatory efficacy, the PEP-1 was selected for the evaluation of its anti-inflammatory potential in an in vivo genetic model of colitis (IL-10KO mice). PEP-1 treatment effectively reduced inflammatory responses and disease severity, with less infiltration of immune cells in the sub-mucosal tissues of the colon in colitic IL-10KO mice.

As aforementioned, GC-up-regulated GILZ plays a key role in GC-mediated NF-κB inhibition and anti-inflammatory activity [44]. Here, we confirmed that GILZ-based PEP-1 peptide inhibited activation and nuclear translocation of the transcription factor NF-κB in colon lamina propria leukocytes, suggesting that it may exert its anti-inflammatory effects by inhibiting the NF-κB pathway. This is consistent with previous studies showing that NF-κB plays a critical role in the pathogenesis of IBD by regulating the expression of pro-inflammatory cytokines and chemokines. Indeed, the nuclear translocation of NF-κB is essential for its biological activity, and molecules that interfere with NF-κB activation and nuclear translocation represent an attractive way to suppress inflammation [56,57,58,59,60]. However, although an increasing number of NF-κB inhibitors have been described, most of these compounds either lack specificity or disrupt the normal physiological functions of NF-κB with consequent unwanted effects so that they are less than ideal candidates for clinical use [58,61].

To further confirm the in vivo efficacy of PEP-1 to counteract colon inflammation, we have also tested it in a different experimental model of acute colitis, chemically induced by intrarectal administration of DNBS [32,48]. PEP-1 treatment was effective in protecting C57BL/6J mice against DNBS-induced colitis, with a significant improvement of the colitis signs in PEP-1-treated mice compared to controls. 

Altogether, these results suggest that PEP-1 peptide may have broad therapeutic potential in treating different types of colitis, acting through inhibition of NF-κB-mediated leukocyte activation. 

## 5. Conclusions

Collectively, these results indicate that PEP-1 GILZ-based peptides possess anti-inflammatory properties like those of GCs and, therefore, can be exploited therapeutically. The overall findings of this study suggest that PEP-1 could serve as a valuable therapeutic agent for the treatment of IBDs. Further investigations are required to assess the safety and efficacy of PEP-1 in human subjects. Nevertheless, this study offers significant insights into the underlying mechanisms of IBDs and highlights the therapeutic potential of targeting the NF-κB pathway for managing these debilitating diseases.

## Figures and Tables

**Figure 1 cells-12-02294-f001:**
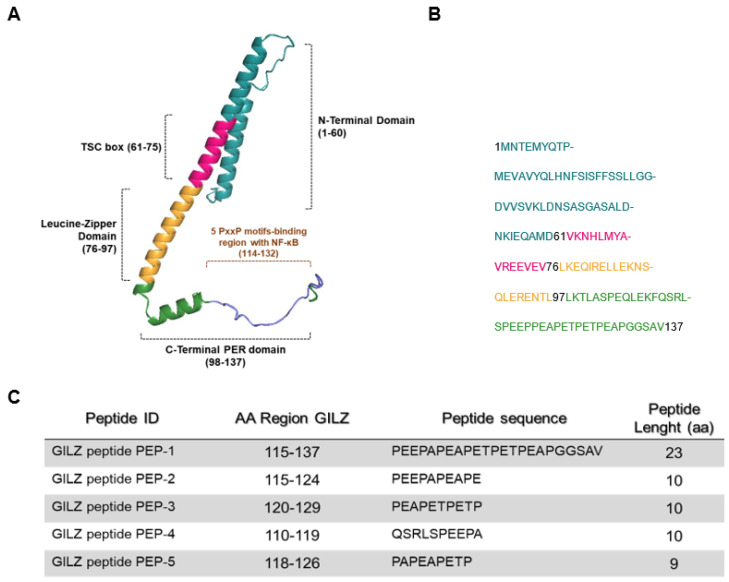
Predictive structure of GILZ protein. (**A**) A three-dimensional representation, created by COSMIC^2^ platform (https://cosmic-cryoem.org, accessed on 15 September 2023) and modified with PyMOL Molecular Graphics System (http://www.pymol.org/pymol accessed on 15 September 2023), of GILZ protein constituted by four domains: a N-terminal domain (NTD, 1–60 aa), TSC-box (61–75 aa), a leucine zipper (LZ, 76–97 aa), and a PER region spanning from 98 to 137 aa residues. (**B**) The primary murine sequence of GILZ protein. (**C**) The table represents sequences of five GILZ peptides spanning different portions of the PER region, which is responsible for GILZ-NF-κB protein-to-protein interaction [21].

**Figure 2 cells-12-02294-f002:**
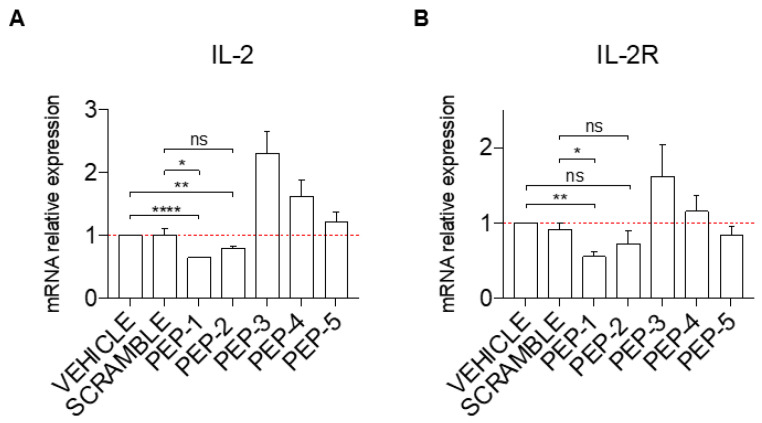
PEP-1 decreases the mRNA expression level of *IL-2* and *IL-2Rα* in Jurkat cells. The level of IL-2 (**A**) and IL-2Rα (**B**) mRNA relative expression was evaluated by qPCR in Jurkat cells treated or not with GILZ peptides (0.1 μM) for 1 h. Afterward, cells were activated with PMA and Ionomycin for 6 h and treated with acetonitrile alone (vehicle), scramble, or PEP peptides (0.1 μM), as indicated in the figure. Graphs represent the mean ± SEM of three independent experiments. * *p* < 0.05, ** *p* < 0.005, **** *p* < 0.0001, *P*-values were calculated according to unpaired Student’s *t*-test. ns: non-significant.

**Figure 3 cells-12-02294-f003:**
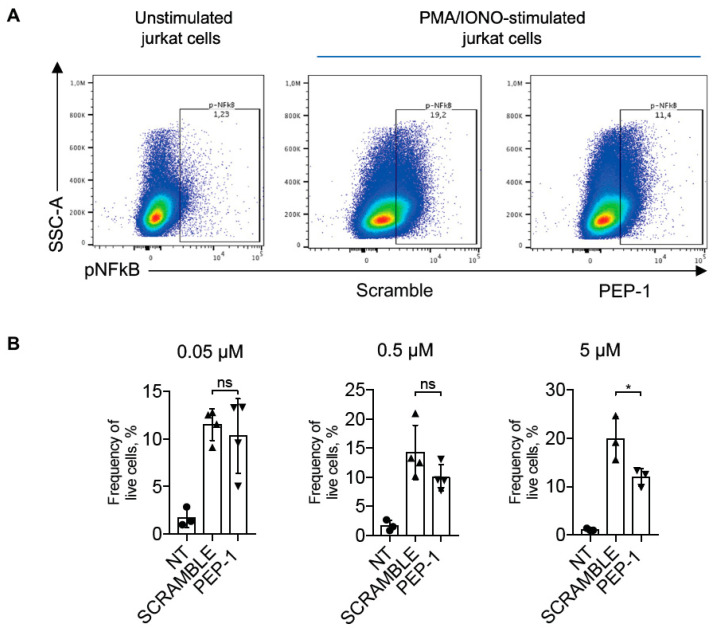
PEP-1 decreases pNF-κB/p65 activation in Jurkat cells. Representative dot plots of flow cytometry analysis of p-NF-κB/p65 in Jurkat cells stimulated or not with PMA and ionomycin for 15 min (**A**). Numbers within quadrants represent the frequency of the gated population of positive pNF-κB cells treated with scramble or PEP-1 peptide (5 μM), compared to the frequency of positive pNF-κB unstimulated Jurkat cells. After that, graphs represent the frequency of pNF-κB+ live Jurkat cells treated with different doses of PEP-1 at the concentration indicated in the above graphs (**B**). Graphs represent the mean of at least three independent experiments ± SD. * *p* < 0.05. *p*-values were calculated using the unpaired Student’s *t*-test. ns: non-significant.

**Figure 4 cells-12-02294-f004:**
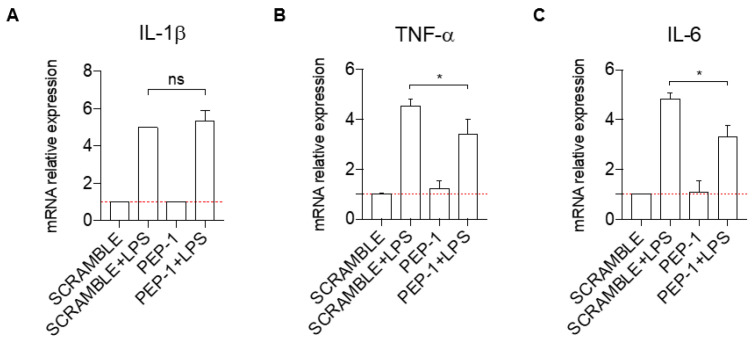
PEP-1 decreases the mRNA expression level of pro-inflammatory cytokines in the THP-1 cell line. IL-1β (**A**), TNF-α (**B**), and IL-6 (**C**) mRNA relative expression was evaluated by qPCR for THP-1 cells activated with LPS, with GILZ peptide PEP-1 (5 μM) for 1 h. Data have been normalized, and activated THP-1 cells treated with the scrambled sequence of PEP-1 peptide were used as control. Graphs represent the mean ± SD of three independent experiments. * *p* < 0.05. *p*-values were calculated using the unpaired Student’s *t*-test. PMA: Phorbol Myristate Acetate. LPS: Lipopolysaccharide.

**Figure 5 cells-12-02294-f005:**
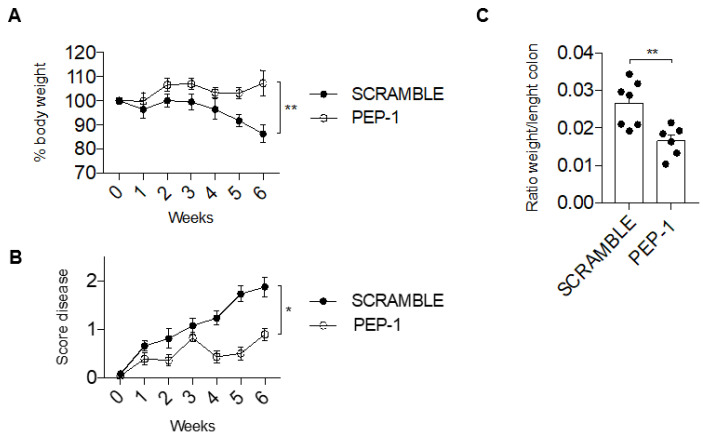
GILZ PEP-1 peptide treatment reduces the clinical signs of colitis in IL-10KO mice. After an intraperitoneal injection of PEP-1, or scrambled sequence (control) at week 7, mice were monitored daily for 6 weeks, and the percentage of body weight loss (**A**) was reported. Graph of total score disease (**B**) was documented, taking into consideration the stool consistency and rectal bleeding. Following the sixth week, mice were sacrificed to calculate the ratio between weight and length of the colon (**C**). Graphs represent mean ± SEM. * *p* < 0.05, ** *p* < 0.005. *p*-values were calculated using the unpaired Student’s *t*-test. Each dot represents an individual mouse (n = 8/group).

**Figure 6 cells-12-02294-f006:**
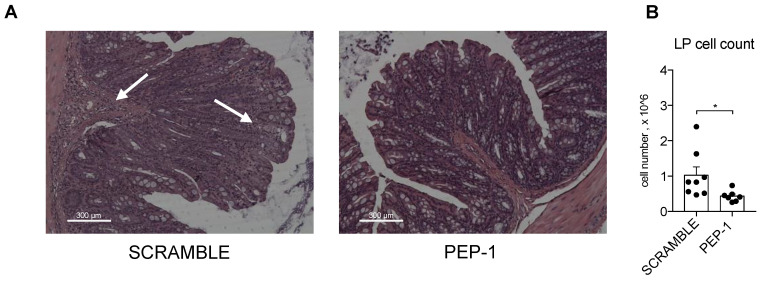
PEP-1 reduces leukocyte infiltration in the colon LP of IL-KO mice. Mice were sacrificed to estimate the infiltration of immune cells in the sub-mucosal tissues through histological examination by H&E staining (**A**). Lamina propria was infiltrated by inflammatory cells (arrowhead) in IL-10KO mice treated with scrambled peptide, unlike the result observed in mice treated with PEP-1. Subsequently, leukocytes infiltrated in LP were isolated, and the cell count was carried out (**B**) in mice treated with scramble or PEP-1. The graph represents mean ± SEM. * *p* < 0.05. *p*-values were calculated using the unpaired Student’s *t*-test. Each dot represents an individual mouse. (n = 8/group).

**Figure 7 cells-12-02294-f007:**
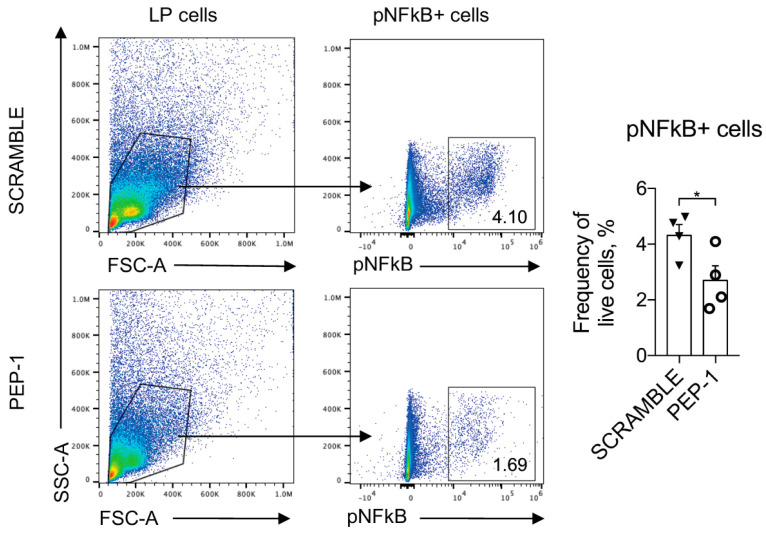
PEP 1 reduces p65/NF-κB phosphorylation in leukocytes infiltrating the colon LP of IL-10KO mice. Representative dot plots of flow cytometry analysis and frequency of cells positively stained for phosphorylated form of p65/NFκB (pNFκB+) among lymphocytes isolated from colon LP of IL-10KO mice treated with scramble or PEP-1 peptide. Numbers within quadrants represent the frequency of the gated population. Graphs represent mean ± SEM. * *p* < 0.05. *p*-values were calculated using the unpaired Student’s *t*-test. Each dot represents an individual mouse. (n = 8/group).

**Figure 8 cells-12-02294-f008:**
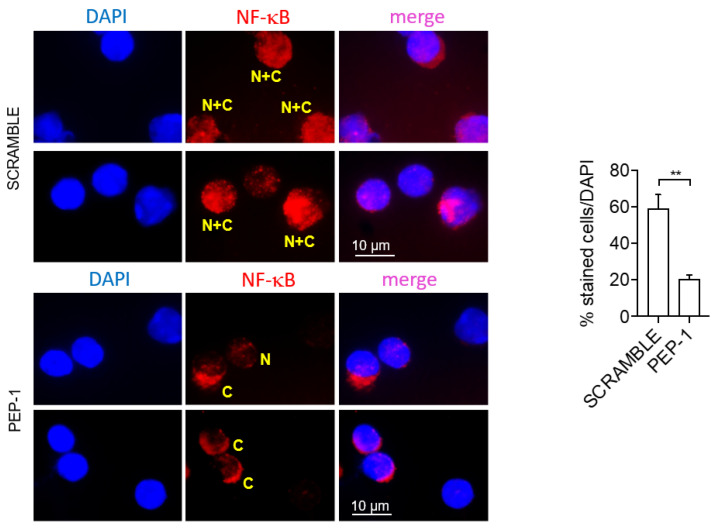
PEP 1 reduces NF-κB nuclear translocation in leukocytes infiltrating the colon LP of IL-10KO mice. Representative immunofluorescence pictures showing pNF-κB/p65 positive cells of isolated colon LP cells from IL-10KO mice; quantification of NF-κB nuclear translocation was calculated by measuring the NF-kB localization to the cytoplasmic region and normalized by counting the total number of cells stained with DAPI. “N + C” (yellow): NF-κB staining in both the nucleus and cytoplasm; “C” (yellow): NF-κB staining in the cytoplasm; “N” (yellow): NF-κB staining in the nucleus. Graph sideways: cell count by ImageJ software of cytoplasmic NF-κB positive cells compared to all DAPI positive cells (representing the whole number of cells present in the slides analyzed). The graph represents mean ± SD. ** *p* < 0.005. *p*-values were calculated using the unpaired Student’s *t*-test. (n = 8/group).

**Figure 9 cells-12-02294-f009:**
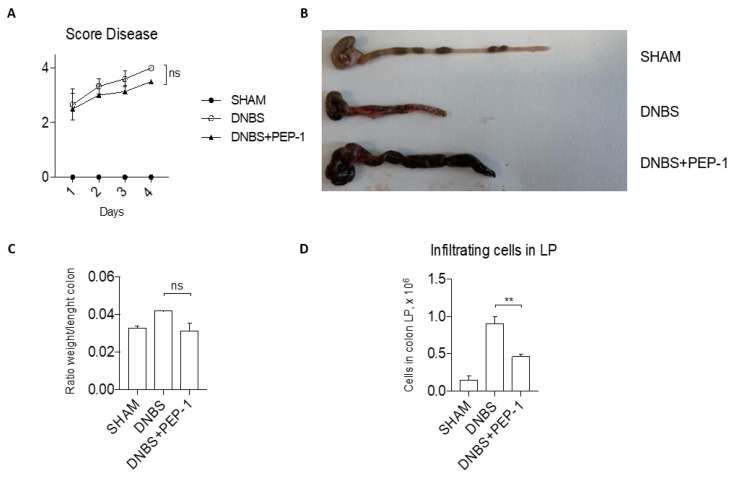
PEP-1 ameliorates clinical signs of colitis in DNBS-induced colitis mice. Before sacrifice, daily measurement of total disease score was conducted in SHAM and PEP-1 treated mice during DNBS-induced colitis (**A**). Following that, colon tissues from mice treated only with vehicle (SHAM), DNBS-induced colitis mice (DNBS), and DNBS-treated mice with PEP-1 injection (DNBS + PEP-1) were examined under a microscope (**B**), and the ratio weight/length of the colon was calculated (**C**); lymphocytes infiltrated in LP were isolated and counted from SHAM, DNBS-induced colitis mice, treated or not with PEP-1 (**D**). Graphs represent mean ± SEM. ** *p* < 0.005. *p*-values were calculated using the unpaired Student’s *t*-test. ns: non-significant. (n = 6/group).

## Data Availability

Data sets generated are available in the current manuscript.

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
