# Peer review of "Anti-Inflammatory Effects of Synthetic Peptides Based on Glucocorticoid-Induced Leucine Zipper (GILZ) Protein for the Treatment of Inflammatory Bowel Diseases (IBDs)"

_cells, 2023, doi:10.3390/cells12182294_

Round 1
Reviewer 1 Report
Comments to the authors:
In this study Paglialunga and colleagues investigated the effect of GILZ peptides on NFkB activation in immune cells and therapeutic effects in two different models of inflammatory bowel disease. This study comes from the discoverers of GILZ and represents an interesting translational study demonstrating glucocorticoid-mediated anti-inflammatory effects can be separated from side-effects by specifically targeting GILZ. The use of GILZ peptides in vivo makes the study very translational and relevant. However, there are various aspects of the study, which are disturbing and require improvement, resp. clarification.
Specific comments:
1) Cell lines such as THP-1 and Jurkat, for proof-of-principle are fine, however, primary cells such as T cells and macrophages would be more relevant. Later in the colitis model the authors use primary cells to analyze NFkB translocation, resp. activation, but gene expression is not analyzed.
2) The cell density in the in vitro experiments is with 1 x 10^6 super high. This may work for short gene expression experiments but is too high for longer culture experiments, e.g. for analyzing cytokine release, which would have been very interesting and improve the study
3) Some studies were reporting that GILZ has a nuclear localization. Thus, why should GILZ, resp. GILZ peptides prevent NFkB nuclear translocation? Overall, the effect of GILZ on NFkB nuclear translocation is not very convincing. the flow cytometry data only analyze p65 phosphorylation, not translocation, and the fluorescence microscopy pictures are not of sufficient quality. In part also because the cytoplasm of primary T cells is very small. In THP-1 cells this would be easier to analyze
4) Figure 2: the authors only report about the inhibitory effect of PEP-1 but not about the much stronger stimulatory effect of PEP-3
5) Figure 3A: axes are not labeled. Overall, the methods part is poorly described and often it is unclear how data were analyzed and presented
6) Although the data are often convincing, statistical analyses appear not be done properly. E.g. in Figure 3B single experiments in triplicates appear to be analyzed, and SEM instead of SD is used.
7) This reviewer is surprise dthat THP-1 as a macrophage cell line produces IL-2. Why was IL-1 beta not analyzed?
8) line 327: in an in vivo setting one is applying a dose of a drug, not a concentration, thus equimolar concentrations cannot be compared
9) In general histologies should be shown and not only disease scores. Also histologies should be compared to untreated or control animals. For example the inflammation in the IL-10 KO mouse shown in figure 6A is not very strong
10) The number of mice analyzed in specific experiments should be indicated
11) It is unclear why certain parts of the results sections are in bold
12) figure 6: the authors state lymphocyte infiltration, yet the method only states leukocyte analysis. How were non-hematopoietic cell distinguished?
13) Figure 8: how many cells were analyzed. Staining and unstimulated controls should be shown as well
Author Response
We thank the reviewers for the positive evaluation of our manuscript and for their suggestions to improve it. We have addressed all the comments, and the Response Letter shows our responses to each point in bold. The uploaded manuscript is a revised version with the introduced changes colored in yellow. We hope that the manuscript is now suitable for publication in Cells journal.
Reviewers’ Comments:
Reviewer #1:
In this study Paglialunga and colleagues investigated the effect of GILZ peptides on NFkB activation in immune cells and therapeutic effects in two different models of inflammatory bowel disease. This study comes from the discoverers of GILZ and represents an interesting translational study demonstrating glucocorticoid-mediated anti-inflammatory effects can be separated from side-effects by specifically targeting GILZ. The use of GILZ peptides in vivo makes the study very translational and relevant. However, there are various aspects of the study, which are disturbing and require improvement, resp. clarification.
Specific comments:
Point 1) Cell lines such as THP-1 and Jurkat, for proof-of-principle are fine, however, primary cells such as T cells and macrophages would be more relevant. Later in the colitis model the authors use primary cells to analyze NFkB translocation, resp. activation, but gene expression is not analyzed.
Response 1: Thank you for this consideration and suggestion; in fact we are planning to perform experiments using primary human cells, however we still do not have an approval from local ethical committee for the use of human primary cells in this study, thus it would not be technically achievable to provide the data within the revision time.
Point 2: The cell density in the in vitro experiments is with 1 x 10^6 super high. This may work for short gene expression experiments but is too high for longer culture experiments, e.g. for analyzing cytokine release, which would have been very interesting and improve the study.
Response 2: We maintained the cells at a density of 1 x 10^6/ml corresponding to the exponential phase of growth in the experiments with short term cultures (6 hours) for the evaluation of cytokine expression by real time qPCR. In fact, much lower starting cell density would be needed for the analysis of the protein expression at a later timepoints by Elisa or IC-FACS, which was not evaluated in the in vitro experimental set up mainly aimed at evaluation of NfKB-mediated transcription.
Point 3: Some studies were reporting that GILZ has a nuclear localization. Thus, why should GILZ, resp. GILZ peptides prevent NFkB nuclear translocation? Overall, the effect of GILZ on NFkB nuclear translocation is not very convincing. the flow cytometry data only analyze p65 phosphorylation, not translocation, and the fluorescence microscopy pictures are not of sufficient quality. In part also because the cytoplasm of primary T cells is very small. In THP-1 cells this would be easier to analyze
Response 3: Thank you for your observation. However we have to point out that we do not completely agree on the following points:
- a) In our hands, GILZ has always been observed as localized to the cytoplasm. Only occasionally it was reported as nuclear. In fact, even if it belongs to the leucine zipper family, GILZ does not have a basic-region DNA binding domain, it does not contain a nuclear localization signal, and there are no works that unequivocally demonstrate its presence in the nucleus. Although GILZ nuclear localization cannot be excluded, so far there are no robust evidence of its presence in the nucleus. To our knowledge, the antibody used to claim its role as a transcription factor (e.g. in ChIP assays) is not specific for GILZ. On the other hand, cytoplasmic localization and interference with other transcription factors in the cytoplasm has been extensively reported by ours and other laboratories using different experimental approaches (PMID 17492054, 20018851, 22110132, 35835370, 34571877). Also, description of GILZ cytoplasmic localization in various tissues is documented in Protein Atlas website at https://www.proteinatlas.org/ENSG00000157514-TSC22D3/tissue;
- b) about the effect of GILZ on NF-kB nuclear translocation other studies have been already published by our and several other groups (PMID: 11468175; 12393603, 18996377, 17169985, 27629417, 21965677, 25876761, 30338290). We modified the description of Figure 8 by addition of nuclear (N) and cytoplasmic (C) labels of NF-κB in the images, and the Figure legend 8 has been corrected accordingly (lines 501-510). Moreover, phosphorylation of p65/NFkB at different sites, including those in position 536, is an accepted prerequisite for the activation and migration of NF-kB to the nucleus (e. as reported in PMID: 11520989, 11239468, 23343355, 25749044). We strongly believe that demonstrating NF-κB cytoplasmic localization in T lymphocytes isolated from lamina propria of the gut in mice treated in vivo with PEP-1 represents the best proof of the efficacy of this GILZ-based peptide.
Point 4) Figure 2: the authors only report about the inhibitory effect of PEP-1 but not about the much stronger stimulatory effect of PEP-3.
Response 4: We thank the reviewer for raising this comment. In the revised manuscript we have more clearly stated the reason of the choice of the PEP-1 for further functional studies, and discussed the opposite results obtained with PEP-3 peptide. This aspect it is now explicitly described in the revised manuscript in result section, lines 292-303, and in the discussion section, lines 620-634.
Point 5) Figure 3A: axes are not labeled. Overall, the methods part is poorly described and often it is unclear how data were analyzed and presented.
Response 5: We have amended Figure 3A and the text of Figure 3 legend (lines 351-358) to be more detailed and exhaustive, as suggested.
Point 6) Although the data are often convincing, statistical analyses appear not be done properly. E.g. in Figure 3B single experiments in triplicates appear to be analyzed, and SEM instead of SD is used.
Response 6: We have now used SD values instead of SEM for statistical analysis and we have amended Figure legend 3, line 357.
Point 7) This reviewer is surprised that THP-1 as a macrophage cell line produces IL-2. Why was IL-1 beta not analyzed?
Response 7: We apologize for the mistake in figures assembling, a panel data of IL-2 of jurkat cells were erroneously loaded in Figure 4A. In the revised Figure 4 we have provided the results for the qPCR analysis of the IL-1beta expression in THP-1 cells, and commented them in the Results section, lines 361-365.
Point 8) line 327: in an in vivo setting one is applying a dose of a drug, not a concentration, thus equimolar concentrations cannot be compared.
Response 8: we have substituted concentration indicated in line 327 with dose, line 391 of the revised version of the manuscript.
Point 9) In general histologies should be shown and not only disease scores. Also histologies should be compared to untreated or control animals. For example the inflammation in the IL-10 KO mouse shown in figure 6A is not very strong.
Response 9: As indicated in the Material and Methods section, paragraph 2.4, presented score disease is a macroscopical index of the evaluation of the disease evolution in the follow up of the experiment, before sacrifice the mice, which does not include evaluation of histological analysis. This way of following the disease takes into account several parameters and is well accepted as a methods to compare disease progression. Surely, the histology often supports the results of the disease score, although is more labor consuming and only visually would add to the information provided with the disease score.
It is true that the inflammation observed in the colon of IL10 KO mice is not very strong, but certainly evident. In this genetic model of colitis there is a certain degree of variability in the onset of the disease and it is not always easy to find the right conditions to document all the parameters of the evolution of the disease, bearing in mind that we have a limited number of animals to use to respect the principle of the 3Rs, that the data obtained were sufficient for an assessment of the level of tissue inflammation.
Point 10) The number of mice analyzed in specific experiments should be indicated
The number of mice has been added in Figure legends 5-9.
Point 11) It is unclear why certain parts of the results sections are in bold
Response 11: We apologise for the formatting mistake, it has now been corrected.
Point 12) figure 6: the authors state lymphocyte infiltration, yet the method only states leukocyte analysis. How were non-hematopoietic cell distinguished?
Response 12: Thank you for your notice; in fact, the Percoll method allow purification of leukocytes and discards non-hematopoietic cells. We have substituted the term lymphocyes with leukocytes in the Figure legend 6 (line 451).
Point 13) Figure 8: how many cells were analyzed. Staining and unstimulated controls should be shown as well
Response 13: We have modified Figure legend 8 (lines 501-510) and uploaded a version of the Figure 8 with the description of Figure 8 by addition of nuclear (N) and cytoplasmic (C) labels of NF-κB in the images.
Reviewer 2 Report
The manuscript by Paglialunga et al. represents a continuation of the authors' long-term investigation for anti-inflammatory effects of glucocorticoid-induced leucine zipper (GILZ) protein. Here, they describe the study of the anti-inflammatory properties of 5 peptides corresponding to the C-end of GILZ protein. The manuscript is written in a very logical way and describe the obtained results in understandable manner. A special importance of the study is that it aims to obtain clinically significant results and in the future may lead to the development of a new anti-inflammatory drug for the treatment of colitis.
However, some shortcomings should still be fixed:
Major points
1. The authors interpret the effect of 5 peptides (designated as PEP-1-5) on the IL-2/IL-2Ra expression levels as "...treatment of cells with PEP-1 and PEP-2 caused a significant reduction in IL-2 mRNA expression levels compared to the controls... On the contrary, activated Jurkat cells treated with PEP-3, PEP-4, and PEP-5 peptides did not show significant differences in IL-2/IL-2Ra expression levels compared to control..." (Figure 2).
Although in fact, all these peptides have the opposite effects on the IL-2 expression - from a decrease of 30-40% (PEP-1) to an increase of 2.5-fold (PEP-3), and these data perfectly correspond to the IL-2Ra expression levels (even not being statistically significant, according the Figure 2).
The authors chose the only one of all these peptides that most corresponds to the initial hypothesis of this study, and focused on its further investigation.
I think it would be more correct to adequately describe the results presented on the Figure 2 and give a specific explanation of why this particular peptide (PEP-1)was chosen for further research - although a comparative analysis of the effects of PEP-1 (a decrease of IL-2 expression by 30-35%) and PEP-3 (an increase of IL-2 expression by 2.5-fold) would give an interesting information of the molecular mechanisms of GILZ effect towards IL-2/IL-2Ra expression.
2. In the light of remark 1, questions arise about some of the wording in the following sections of the text. The phrase "...Based on the in vitro data acquired, it was observed that the peptide PEP-1 exhibited the highest efficacy..." (lines 320-321) looks incorrect and needs to be changed.
3. Figure 5A - according to the signature to the Y axis, by week 6, the PEP-treated animals lost about 100% of their weight (more than control ones) - and this is in principle impossible. Most likely, it is a technical oversight for this Figure which should be corrected.
4. Authors investigated the same parameter (ratio weight/length for colon) in two different mouse model - IL-10 Ko mice (strain?) and C57BL/6JOlaHsd mice. And according the presented results, IL-10 KO mice have 5-times higher weight/colon ratio (0,18-0,26) compared with C57BL/6JOlaHsd mice (0,036-0,04). This needs an explanation.
5. Discussion of the results obtained is practically absent - this section should be written.
a) The first paragraph of the Discussion section (lines 420-448) represents introductory information, which need to be moved (and appropriately combined) to the Introduction section.
b) In the next paragraph of the Discussion, the authors repeat the aim of this study with some impotant technical information - that should be moved into the Materials and Methods/Results section.
c) The subsequent paragraphs represent repeating of the results presented in the manuscript, and the only reference to the published data is a reference to the previous papers by the authors itself (refs 51-54, lines 474-478).
Minor points
6. There are unfortunate expressions in the manuscript text that need to be corrected, for example
"... of the mRNA expression of IL-2 expression..." (lines 306-307).
7. It is also necessary to correct the typos and formatting of the manuscript (lines 320-336).
Moderate editing of English language required
Author Response
We thank the reviewers for the positive evaluation of our manuscript and for their suggestions to improve it. We have addressed all the comments, and the Response Letter shows our responses to each point in bold. The uploaded manuscript is a revised version with the introduced changes colored in yellow. We hope that the manuscript is now suitable for publication in Cells journal.
Reviewers’ Comments:
Reviewer #2:
The manuscript by Paglialunga et al. represents a continuation of the authors' long-term investigation for anti-inflammatory effects of glucocorticoid-induced leucine zipper (GILZ) protein. Here, they describe the study of the anti-inflammatory properties of 5 peptides corresponding to the C-end of GILZ protein. The manuscript is written in a very logical way and describe the obtained results in understandable manner. A special importance of the study is that it aims to obtain clinically significant results and in the future may lead to the development of a new anti-inflammatory drug for the treatment of colitis.
However, some shortcomings should still be fixed:
Major points
Point 1: The authors interpret the effect of 5 peptides (designated as PEP-1-5) on the IL-2/IL-2Ra expression levels as "...treatment of cells with PEP-1 and PEP-2 caused a significant reduction in IL-2 mRNA expression levels compared to the controls... On the contrary, activated Jurkat cells treated with PEP-3, PEP-4, and PEP-5 peptides did not show significant differences in IL-2/IL-2Ra expression levels compared to control..." (Figure 2).
Although in fact, all these peptides have the opposite effects on the IL-2 expression - from a decrease of 30-40% (PEP-1) to an increase of 2.5-fold (PEP-3), and these data perfectly correspond to the IL-2Ra expression levels (even not being statistically significant, according the Figure 2).
The authors chose the only one of all these peptides that most corresponds to the initial hypothesis of this study, and focused on its further investigation.
I think it would be more correct to adequately describe the results presented on the Figure 2 and give a specific explanation of why this particular peptide (PEP-1)was chosen for further research - although a comparative analysis of the effects of PEP-1 (a decrease of IL-2 expression by 30-35%) and PEP-3 (an increase of IL-2 expression by 2.5-fold) would give an interesting information of the molecular mechanisms of GILZ effect towards IL-2/IL-2Ra expression.
Response 1: We would like to thank the reviewer for raising this question. We have modified the Results, lines 292-303, and in Discussion section, lines 620-634, to better highlight the results obtained with different peptides.
2. In the light of remark 1, questions arise about some of the wording in the following sections of the text. The phrase "...Based on the in vitro data acquired, it was observed that the peptide PEP-1 exhibited the highest efficacy..." (lines 320-321) looks incorrect and needs to be changed.
Response 2: We agree with the reviewer and we have amended the phrase with “… peptide PEP-1 exhibited the most significant inhibitory effect on pro-inflammatory cytokine IL-2 transcription”, lines 386-387 in the revised manuscript.
3. Figure 5A - according to the signature to the Y axis, by week 6, the PEP-treated animals lost about 100% of their weight (more than control ones) - and this is in principle impossible. Most likely, it is a technical oversight for this Figure which should be corrected.
Response 3: We apologize for this mistake, Y axis in figure 5A has been corrected.
4. Authors investigated the same parameter (ratio weight/length for colon) in two different mouse model - IL-10 Ko mice (strain?) and C57BL/6JOlaHsd mice. And according the presented results, IL-10 KO mice have 5-times higher weight/colon ratio (0,18-0,26) compared with C57BL/6JOlaHsd mice (0,036-0,04). This needs an explanation.
Response 4: We thank the reviewer for noticing this discrepancy, a mistake occurred in presenting the weight/length ratio in Figure 5C, the correct range of weight/colon ratio is 0,018/0,026 and not 0,18-0,26; it has been corrected accordingly in Figure 5C.
Point 5: Discussion of the results obtained is practically absent - this section should be written.
a) The first paragraph of the Discussion section (lines 420-448) represents introductory information, which need to be moved (and appropriately combined) to the Introduction section.
b) In the next paragraph of the Discussion, the authors repeat the aim of this study with some impotant technical information - that should be moved into the Materials and Methods/Results section.
c) The subsequent paragraphs represent repeating of the results presented in the manuscript, and the only reference to the published data is a reference to the previous papers by the authors itself (refs 51-54, lines 474-478).
Response 5: A new version of the Discussion substitutes the older one in the revised version of the manuscript (lines 553-661). As suggested, parts of the discussion has been moved to Introduction, Materials and Methods and Results sections (highlighted in yellow in the revised manuscript). Six new references have been added in the revised version of the manuscript.
Minor points
Point 6: There are unfortunate expressions in the manuscript text that need to be corrected, for example
"... of the mRNA expression of IL-2 expression..." (lines 306-307).
Response 6: We thank the reviewer for noticing it, it has been corrected.
Point 7. It is also necessary to correct the typos and formatting of the manuscript (lines 320-336).
Response 7: Thank you for pointing out the errors. They have been corrected in the revised version of the manuscript.
Round 2
Reviewer 1 Report
The authors have convincingly answered all my questions